# Monitoring and Assessment of Rehabilitation Progress on Range of Motion After Total Knee Replacement by Sensor-Based System

**DOI:** 10.3390/s20061703

**Published:** 2020-03-18

**Authors:** Yo-Ping Huang, Yu-Yu Liu, Wei-Hsiu Hsu, Li-Ju Lai, Mel S. Lee

**Affiliations:** 1Department of Electrical Engineering, National Taipei University of Technology, Taipei 10608, Taiwan; raimen1992@gmail.com; 2Department of Computer Science and Information Engineering, National Taipei University, New Taipei City 23741, Taiwan; 3Department of Orthopedic Surgery, Chang Gung Memorial Hospital, Chiayi 61363, Taiwan; 7572@adm.cgmh.org.tw; 4Department of Ophthalmology, Chang Gung Memorial Hospital, Chiayi 61363, Taiwan; lynnlai@kimo.com; 5Department of Orthopedic Surgery, Chang Gung Memorial Hospital, Kaohsiung 83301, Taiwan; mellee@cgmh.org.tw

**Keywords:** inertial measurement unit (IMU), rehabilitation progress, range of motion (ROM), total knee replacement (TKR)

## Abstract

For total knee replacement (TKR) patients, rehabilitation after the surgery is key to regaining mobility. This study proposes a sensor-based system for effectively monitoring rehabilitation progress after TKR. The system comprises a hardware module consisting of the triaxial accelerometer and gyroscope, a microcontroller, and a Bluetooth module, and a software app for monitoring the motion of the knee joint. Three indices, namely the number of swings, the maximum knee flexion angle, and the duration of practice each time, were used as metrics to measure the knee rehabilitation progress. The proposed sensor device has advantages such as usability without spatiotemporal constraints and accuracy in monitoring the rehabilitation progress. The performance of the proposed system was compared with the measured range of motion of the Cybex isokinetic dynamometer (or Cybex) professional rehabilitation equipment, and the results revealed that the average absolute errors of the measured angles were between 1.65° and 3.27° for the TKR subjects, depending on the swing speed. Experimental results verified that the proposed system is effective and comparable with the professional equipment.

## 1. Introduction

A survey revealed that more than 80% of elderly people aged 75 years or older were encumbered by degenerative arthritis [1]. Furthermore, another study reported that one of every 100 people had rheumatoid arthritis [2]. According to a report by the Agency for Healthcare Research and Quality, more than 600,000 Americans undergo total knee replacement (TKR) every year, while more than 15,000 people undergo TKR in Taiwan each year [3,4]. Osteoarthritis is the most common reason for knee replacement operation in the USA [3]. The success of the replacement highly depends on the presurgery assessment and continuing post-surgery rehabilitation. Physical therapy is an essential part of rehabilitation after TKR. Regular rehabilitation is crucial for patients to adapt to the replaced artificial joint and gradually regain their mobility [5].

According to a hospital’s tracking report, some patients after TKR had knees that remained swollen or even deteriorated when the patients returned to the hospital for inpatient services [4]. Whether the surgical procedure was not performed properly and whether the patients lacked proper rehabilitation after returning home for recuperation are important clinical issues that concern orthopedists [6]. Therefore, in this paper, we propose an effective method that integrates a designed sensor device, an inertial measurement unit (IMU), and an Android smartphone for monitoring the progress of rehabilitation after TKR. Furthermore, a software app was designed for long-term monitoring of the effect of rehabilitation by orthopedists and patients.

With the advancement of electronic technologies and their integration with wearable devices, the underlying systems have been extensively applied in the field of medicine, especially for tracking rehabilitation actions. To assess the evidence of wearable devices supporting their efficacy in assisting rehabilitation after total hip replacement and TKR, a review by Bahadori et al. [7] found that both accelerometer and gyroscope were used in five studies. To objectively assess the lengthy process of rehabilitation for cerebrovascular diseases (e.g., stroke), Friedman et al. [8] used a wearable sensor device (magnetometer) to precisely monitor daily use of the wrists and fingers. Mariani et al. [9] attached wireless inertial sensors on shoes to estimate the heel and toe clearance for gait analysis. They proposed independent 2-D and 3-D models for motion detection and three other models for estimating gait rehabilitation information from patients without them being confined to a specific location and experimental environment for data collection.

Inertial measurement units (IMUs) have been widely applied to detect human postures and gaits in recent years. As a result of their low cost, ease of use, and low weight, sensors can be easily integrated with wearable devices and the Internet of things for various applications. IMUs can be used in posture identification for applications in sports and healthcare.

Van Der Straaten et al. [10] gave systematic review to investigate the application of inertial sensor systems and kinematics obtained from systems in their study and to provide assessment to people with knee osteoarthritis and TKR. Kontadakis et al. [11] introduced a gamified rehabilitation platform consisting of a mobile game and an IMU placed on a lower limb in order to capture its orientation in space in real-time for patients undergone TKR. Jones et al. [6] attached IMUs on the lower limbs to objectively distinguish four rehabilitation exercises that were prescribed to osteoarthritis patients following TKR. Seel et al. [12] used an IMU for joint axis and position identification, and for flexion/extension joint angle measurement. Bakhshi et al. [13] developed two IMUs that were mounted on the upper leg and the lower leg to measure the knee joint angle.

Evaluation of gait and body motion disorder is relevant to fall risk assessment after knee-joint surgery [14]. Huang et al. [14] used three accelerometers (attached to both wrists and the chest) and applied the signal magnitude vector (SMV) and a type-2 fuzzy system for fall detection. Hsu et al. [15] presented an automatic gait analysis algorithm that can automatically obtain acceleration and angular velocity by using one accelerometer and two gyroscopes. Triaxial accelerometers were used to detect falls events [16,17]. They presented a peak-value detection algorithm that can effectively discriminate the start and end times of each gait sequence. Furthermore, they proposed an algorithm that can improve the detection accuracy and quantify walking behavior under irregular movement. Mei et al. [18] used acceleration sensor signals to determine gait event. Teufl et al. [19] reported step width measurement based on IMUs and achieved valid results for 3D gait analysis. Müller et al. [20] used a Kinect sensor to evaluate body motion and assess gait. Lorenzi et al. [21] presented a mobile healthcare device to monitor human motion disorders. Gholami et al. [22] designed a Kinect system to assess gait parameters in multiple sclerosis patients. Kun et al. [23] developed a sensor set and an algorithm to estimate knee-joint kinematics. However, the captured signals from the aforementioned experiments [23] cannot be processed online.

Data analysis and motion classification based on posture detection can help researchers more clearly understand users’ motion behavior. Some well-known methods such as neural networks, fuzzy modeling, and data mining have been proposed for analyzing motion patterns. Milosevic et al. [24] used self-organizing maps for visualizing trunk muscle synergies during sitting perturbations. Han et al. [25] used a single IMU to detect normal and abnormal gait phases. Hanakova et al. [26] evaluated complex movements of the arm during walking based on gyroscope data and an angle–angle diagram. They also compared the results with those of the range of motion (ROM) method.

From the literature survey, we discovered that there remained problems to be solved in previous studies on rehabilitation monitoring after TKR. Therefore, the developed sensor-based system is aimed to fulfill three objectives: (1) monitoring whether the TKR patients followed the orthopedist’s rehabilitation instructions at home, (2) recording the duration of each rehabilitation session, and (3) determining the extent to which a patient’s knee can flex in each rehabilitation course. To fulfill these objectives, the developed sensor devices had to be reliable, user friendly, and easy to use and had to enable TKR patients to achieve rehabilitation effects at home similar to those achieved using professional equipment in a hospital.

This paper is organized as follows. In Section 2, the proposed method and hardware design are described, and the procedure for calculating the equivalent range of motion (ROM) is detailed. The way to monitor rehabilitation progress is illustrated in Section 3. The experimental results and discussion are provided in Section 4. The conclusions and future work are presented in the Section 5.

## 2. Methodology

In the proposed method, after TKR, patients must follow the orthopedist’s or physical therapist’s instructions to strengthen the knee as well as to avoid complications or dislocation of the new joint. Patients may feel uncomfortable while walking or exercising, and their legs and feet may be swollen. Physical therapists will instruct patients to use professional equipment such as the Cybex to restore movement in the knee and leg. The patient’s leg is bound to the arm of the equipment for rehabilitation. On the basis of the wound recovery progress, the arm’s swing speed is adjusted to the optimal speed suitable for the patient’s current knee condition. The equipment automatically records the ROM, indicating the flexion angle of the knee joint, as shown in Figure 1. The performance of the developed sensor device was validated with respect to Cybex.

### 2.1. Hardware Design

The structure of the proposed system comprises two parts, rehabilitation devices and user information, as shown in Figure 2. The rehabilitation devices were hardware comprising the MPU6050 triaxial accelerometer and gyroscope, a microcontroller, and a Bluetooth module for measuring the knee joint motion. The user information part is an Android-based app designed for displaying rehabilitation-related information for patients.

The developed sensor device is shown in Figure 3. It comprises an ATMEGA328 microcontroller, a MPU6050 module, an Arduino Bluetooth module, a lithium battery (9 V, 650 mAh), and a smartphone. The smartphone is used to receive signals transmitted by the Bluetooth module from the accelerometer and gyroscope. The size of the developed device is 27 mm × 27 mm × 13 mm. To conveniently wear the device on the leg, the device is plugged into a 3D-printed shell 33 mm × 32.5 mm × 16 mm in size. It takes less than 5 sec for a user to wear it. The device can be used up to 17.5 h without recharging. If a patient needs to rehabilitate around an hour per day, the devices can last for at least half a month without recharging. The specifications of the developed sensor device are listed in Table 1. The developed app displays the calculated swing angles to the users on the smartphone. Two sensor devices are worn on the thigh (sensor 1) and ankle (sensor 2), respectively, as shown in Figure 4. In this study, a pair of sensor devices were used for measuring the swing angles for the static mode (patients sitting on the chair for rehabilitation).

### 2.2. System Flowchart

When the users are ready for rehabilitation, they simply must turn on the devices and select the app to receive signals transmitted from the devices. The system flowchart is presented in Figure 5. To obtain accurate swing angles, we apply the signals from both the accelerometer and gyroscope to a Kalman filter [27,28] for smoothing and using the quaternion to calculate the angles. To quantify the rehabilitation angles in the equivalent ROM, the effect of gravity is removed from the accelerometer so that the real acceleration signals from rehabilitation can be obtained [29]. Once the smartphone receives the transmitted signals, it calculates the angles, performs filtering, and quantifies the swings. The filtering function is used to remove small angles, such as 5°, so that they are not considered as effective swing angles. The number and duration of swings in each rehabilitation course are saved in the system.

### 2.3. Effect of Gravity on Angle Measurement

When the swing motion is fast, the measured acceleration from the triaxial accelerometer is influenced by the gravity effect and cannot reflect the actual flexion angle of the leg. To realize the effect of gravity on the measurement, some preliminary experiments were performed to remove the gravity effect. When a user sits on a chair flexing the leg at 90° at different velocities, as shown in Figure 4, the swing angles should be located near 90°. However, the detected angles normally ranged between 70°–110°, as shown in Figure 6. The errors were induced by the effect of gravity. Thus, the acceleration signals cannot be directly applied to calculate the swing angles without removing the effect of gravity.

To remove the effect of gravity, we adopted a method used in fall detection that involves applying the SMV [14], which is expressed as follows:(1)$$SMV\left(t\right)=\sqrt{a_{x}{\left(t\right)}^{2}+a_{y}{\left(t\right)}^{2}+a_{z}{\left(t\right)}^{2}}$$
where *a_x_*(t), *a_y_*(t), and *a_z_*(t) represent the acceleration values from the *X*-, *Y*-, and *Z*-axes at time *t*, respectively. The angles between the SMV and the three axes are denoted by *ρ*, *φ*, and *θ*, respectively, as illustrated in Figure 7.

When the triaxial accelerometer detects an object’s motion, it measures forces exerted by the object as well as by gravity. The total force can be expressed as follows:(2)$$A=G+B\Rightarrow \left[\begin{array}{c} a_{x}\\ a_{y}\\ a_{z} \end{array}\right]=\left[\begin{array}{c} G_{x}+B_{x}\\ G_{y}+B_{y}\\ G_{z}+B_{z} \end{array}\right]$$
where vector *A* is the sum of vectors *G* and *B*, and $A=\left[\begin{array}{c} a_{x}\\ a_{y}\\ a_{z} \end{array}\right]$ is the measured acceleration, $G=\left[\begin{array}{c} G_{x}\\ G_{y}\\ G_{z} \end{array}\right]$ is the acceleration due to gravity on the three axes, and $B=\left[\begin{array}{c} B_{x}\\ B_{y}\\ B_{z} \end{array}\right]$ represents the external force that is caused by the object’s motion. From Equation (2), if we can find the component of gravity on each axis, we can calculate the desired force caused by the object, assuming that acceleration can instantly change with motion, but an object cannot instantly change its orientation. Thus, vector A is oriented in almost the same direction as vector G. Therefore, the projection of vector G on the three axes is expressed as follows:(3)$$G_{x}\approx \|G\|\cdot \cos\left(\rho \right)\;G_{y}\approx \|G\|\cdot \cos\left(\varphi \right)\;G_{z}\approx \|G\|\cdot \cos\left(\theta \right)$$

The angles *ρ*, *ϕ*, and *θ* can be calculated as follows:(4)$$\rho =\cos^{-1}\left(\frac{a_{x}}{\sqrt{a_{x}{}^{2}+a_{y}{}^{2}+a_{z}{}^{2}}}\right)\;\varphi =\cos^{-1}\left(\frac{a_{y}}{\sqrt{a_{x}{}^{2}+a_{y}{}^{2}+a_{z}{}^{2}}}\right)\;\theta =\cos^{-1}\left(\frac{a_{z}}{\sqrt{a_{x}{}^{2}+a_{y}{}^{2}+a_{z}{}^{2}}}\right).$$

From Equations (2) and (3), we can find the desired forces *B_x_*, *B_y_*, and *B_z_*. The results from another experiment after removing the effect of gravity are shown in Figure 8. Most of the swing angles were located near 90°. The error was approximately 10°, which is a considerable improvement compared with that in Figure 6. Some deviations of more than 10° from the right angle occurred because the leg did not return exactly to 90°. This experiment validated that the removal of gravity can reduce its effect on the calculation of the acceleration of an object.

### 2.4. Knee Angle Calculation

This study used a six-axis IMU, which comprises a triaxial accelerometer and triaxial gyroscope [28,30]. The triaxial gyroscope has high accuracy in a short period and sensitivity to motion. However, it exhibits an accumulated error after a long period of measurement, which affects its accuracy. By contrast, the triaxial accelerometer is advantageous because its acceleration is sensitive to change with motion and it has stable measurement accuracy in the long term. We performed Kalman filtering on the rotation angle calculated from the acceleration and the angle integrated from the gyroscope angular velocity to determine the real roll and pitch angles. The rotation angles were calculated from the accelerometer measurements as follows:(5)$$Roll=tan^{-1}\left(\frac{a_{Z}}{a_{Y}}\right)$$
(6)$$Pitch=tan^{-1}\left(\frac{a_{X}}{a_{Z}}\right).$$

The directions of the rotation angles are expressed as shown on the right side of Figure 5. The *Z*-axis was perpendicular to the ground, therefore it was used as the reference direction for acceleration, which made the rotation direction parallel to the *Z*-axis. When the *X*- and *Y*-axes are rotating with the *Z*-axis, the measured acceleration should be (0, 0, +1 g). However, this would make determining the yaw angle impossible. To determine the yaw angle, we applied the quaternion calculation.

The quaternion can be expressed as *q = w + xi + yj + zk*, where (*w*, *x*, *y*, *z*) are the values of the quaternion. When applied to the six-axis inertia module, the quaternion can be calculated from the acceleration and angular velocity. This study used the MPU6050 inertia sensing module, which includes the quaternion in its functional library. We express the function for retrieving the values of the quaternion to calculate the yaw angle as follows:(7)$$Yaw=tan^{-1}\left(\frac{2\left(wz+xy\right)}{1-2\left(y^{2}+z^{2}\right)}\right).$$

The yaw angles calculated using Equation (7) were not smooth. The raw signals needed to be processed by the Kalman filter to obtain the desired signals.

## 3. Monitoring Rehabilitation Progress

During a rehabilitation course, the users’ knee motion angle varies over time. The distribution of the swing angles may range, for example, from 60° to 180°. Even if we were to record all motion angles, there is no simple method to quantitatively measure the effect in each rehabilitation course. To resolve this problem, we applied Fuzzy c-means (FCM) to identify the centroid of the acceleration signals so that an equivalent ROM can be calculated to represent the effect of a rehabilitation course.

### 3.1. Equivalent Angles of Knee Motion from FCM

FCM is one of the commonly used machine learning methods that can softly partition data into the predetermined number of clusters [31]. A datum can be classified into any of the clusters with a membership degree between 0 and 1 under the constraint that the sum of membership degrees should be equal to 1. FCM was applied to calculate the equivalent ROM for the swing angles of rehabilitation. When a pair of the developed sensor devices were worn on the thigh and ankle, as shown in Figure 9, the angle between the thigh and the shank was 180° – *θ*, where *θ* is the angle between the shank and the ground.

The triaxial accelerometer is highly sensitive, therefore any vibration or other disturbance during the rehabilitation course causes an abnormal reading from the device. Therefore, before determining the equivalent angles, gravity was removed and the Kalman filter was then applied to preprocess the received signals from the sensor devices. Using the Kalman filter to preprocess the raw acceleration signals can suppress disturbances and smooth them for further analysis. The equivalent ROM was calculated from the centroids G1 and G2, as shown in Figure 10. The steps to calculate the equivalent swing angles are listed as follows:

Step 1: Record the signals from both sensors 1 and 2. The swing effects from the left and right directions were discarded, and only the acceleration signals from X_g_ and Y_g_ were considered.

Step 2: Use FCM to cluster the signals from each sensor into three groups, represented by fcm_1i_ and fcm_2j_ for *i*, *j* = 1, 2, 3.

Step 3: Find the centroid from each sensor device and represent the pair as (G1, G2).

Step 4: Perform basic operations on inverse trigonometric functions to calculate the equivalent ROM from the centroid pair (G1, G2) as follows:(8)$$\theta_{G1}=\tan^{-1}\left|\frac{-Y_{1}}{X_{1}}\right|.$$
(9)$$\theta_{G2}=\tan^{-1}\left|\frac{-Y_{2}}{X_{2}}\right|.$$
(10)$$EquiROM=\theta_{G2}-\theta_{G1}.$$

### 3.2. Monitoring the Effect of Rehabilitation

An Android smartphone was used to receive and record signals transmitted from the developed sensor devices. The users needed to input some basic information, such as name, age, gender, and the preferred animation type (boat, cow, or car) at the first time of use. The designs for the animation types were based on the fact that most elderly people living in the vicinity of the hospital were fishermen, farmers, and retirees. After this information was input, the smartphone was paired with the sensor devices via Bluetooth, as shown in the lower part of the start page. Once paired successfully, the smartphone was ready to receive and analyze the signals transmitted from the sensor devices. An orthopedist can simultaneously track and monitor a patient’s rehabilitation status. If a patient does not follow the instructions, the orthopedist can actively contact the patient to determine what the problems are.

The maximum swing angle is displayed sequentially in the lower part of the smartphone screen each time, as shown in Figure 11. Properly recording the swing time is crucial for clustering the angles and evaluating whether regular rehabilitation is performed. The counter for each rehabilitation course is highly useful for monitoring recovery progress, as the orthopedist can refer to the counter to decide whether to adjust the rehabilitation course. The counter for each rehabilitation course is also displayed on the screen. We use a color bar to display the percentage of completion of the designated course. The selected animation type, for example a car, moves upon each swing. This approach encourages the patient to continue exercising; otherwise, the car stops moving.

To monitor the effect of rehabilitation, a line chart is used to display each swing angle on the screen, which represents a considerable improvement over the current rehabilitation system in which patients must wait before receiving an examination report. Furthermore, quantile plots are presented to display the swing angles in ascending order as well as to quantitatively compare whether there is noticeable progress after the designated course.

### 3.3. Quantile Plot for Swing Progress

To quantitatively monitor the long-term progress of rehabilitation, we used a quantile plot to display the swing angles on the app. After the completion of one exercise course, the swing angles were saved in the memory and displayed as a line plot in the app, as shown in Figure 12a. The lower part of the screen in Figure 12a showed that the subject performed 52 swings with an average angle of 82.11°. For validating the effect on rehabilitation, the swing angles were rearranged in ascending order. For illustration, two dotted line plots are compared in Figure 12b, where the *X*-axis represents the percentage of swing angles, and the Y-axis represents the corresponding angles. A line drawn perpendicular to 50% (f = 0.5) intersects with the bottom curve at an angle of 60° and with the upper curve at 75°. This implies that the first 50% of the swing angles were lower than 60° in the first rehabilitation course, whereas the leg flexion angle improved to 75° in the second rehabilitation course, indicating progress. By contrast, the swing angle remaining low, the line plots being flat, or the plots not having noticeable differences relative to the previous plots for a long time indicates either that the surgery was not completely successful or that the replaced knee joint gradually hardened. This may require immediate intervention by orthopedists to identify the problems.

## 4. Experimental Results and Analysis

From June 2015 to May 2016, 35 subjects (11 without using Cybex and 24 using Cybex) aged 20–85 years were enrolled for the experiments at Chang Gung Memorial Hospital, Chiayi Branch, Taiwan. Written informed consent was obtained from all subjects. This study was approved by the Institutional Review Board of the Chang Gung Foundation (IRB-104-3347B). The research was conducted according to the principles of the Declaration of Helsinki. A detailed personal history review, general health examination, and lifestyle questionnaire were administered for all subjects.

### 4.1. Results from Subjects Without Using Cybex

Before using the hospital equipment, 11 participants were invited to test the effectiveness of the developed sensor devices in the laboratory. The characteristics of the 11 participants are listed in Table 2. Two sensors were worn on the thigh and ankle, as shown in Figure 4. Each participant sat on a chair and flexed the leg back and forth at approximately 90° 20 times. The swing angles and number of swings from one of the participants are shown in Figure 13. The outcome showed no error in counting the swings from all 11 participants. This validated the accuracy of the developed sensor device.

### 4.2. Equivalent ROM

Cybex is a motorized equipment used in the hospital and its swing speed can be manually adjusted to meet the user’s need. Experiments were then performed using professional rehabilitation equipment, Cybex, as a benchmark. To validate the effectiveness of the proposed system, only one sensor had to be worn at the shank during swings driven by Cybex, as shown in Figure 14. The patient’s leg was bound tightly to the arm of Cybex so that the precise ROM could be recorded. When sensor 1 was worn on the thigh and sensor 2 was worn on the ankle, sensor 1 did not have any effect on the calculation of the ROM because it remained still on the chair and did not have freedom of movement. Although we tested two sensors, there was no noticeable difference in the experimental results. However, two sensor devices were required for calculating the equivalent ROM in home-based rehabilitation. The smaller the difference between the detected angle and the ROM of Cybex, the higher the accuracy of the proposed system is.

There were 16 healthy control subjects, namely eight men aged 59–66 years, six women aged 20–30 years, one woman aged 53 years, and one woman aged 65 years. Table 3 shows the control subjects’ characteristics. Table 4 shows the eight TKR subjects’ characteristics, where TKR (month) in the last row represents the time after surgery.

Each subject wore one sensor device on the right shank and participated in experiments at angular speeds of 25°/s, 60°/s, and 180°/s for swings driven by Cybex. The three controlled speeds corresponded to normal walking speed, running speed, and athlete speed, respectively. The experiment was repeated five times at each angular speed. Then, the subjects repeated the process for the left leg. Only the maximum angle was saved in the database in each swing. Thus, ten data were collected from each subject on both legs at each angular velocity. The experiments were designed to verify the accuracy of the sensor devices with reference to Cybex as the benchmark.

The proposed method was used to calculate the equivalent ROM. The subjects sat on a chair, therefore only the lower limb swung with the equipment. On the basis of the distribution of sensor signals from the accelerometers, where the Y-axis signals were located in the proximity of 0, we could identify that sensor 1 was worn on the thigh and sensor 2 was worn on the ankle. For the example of one experiment course shown in Figure 10, the centroid based on three cluster centers from FCM can be determined to be G1 = (10.263,−0.152) and G2 = (1.263,−9.64) for sensor 1 and sensor 2, respectively. According to Equations (4)–(6), we can calculate the equivalent ROM as 81.687°. For calculating the rest of the angles, we followed the procedures stated here.

### 4.3. Results from Healthy Control Subjects

In total, 160 swing angles should have been received for each angular speed from the 16 healthy control subjects. However, in some experiments, sensor data were not received successfully due to some unexpected situations such as the sensor not being worn tightly and sliding during swings. After these abnormal data were removed, 135 swing angles were available at each angular speed. The ROM is plotted in Figure 15a–c for the three speeds, respectively. The average absolute ROM errors with respect to the Cybex reference equipment were 2.90°, 3.51°, and 4.00° at the three different angular speeds, as shown in Table 5. Although the average absolute errors were acceptable from the orthopedist’s viewpoint, the results showed that the average absolute errors (accuracy) increased (decreased) with the angular speed of Cybex. This can be attributed to two reasons: The first was missed capturing of the maximum swing angle from the Cybex equipment because the sensed signals needed to be transmitted from the device to the smartphone via Bluetooth. The current sampling rate from the sensor device was 100 Hz. There was a trade-off between the sampling rate and the overhead on the smartphone memory. At the instance when the Cybex maximum angle was not captured exactly by the device, calculation error occurred. The second reason was vibration from the subject’s leg. The leg was bound tightly to the Cybex arm, therefore subjects sometimes consciously resisted the swing, causing vibration at the turning point of the maximum swing. The vibration may also induce errors in the sensor devices because they are sensitive to any movement.

We also calculated the correlation coefficients between the two measurement systems under three angular speeds, 25°/s, 60°/s, and 180°/s shown in Figure 14, to be 0.975, 0.969, and 0.967, respectively. These coefficients indicated high consistency between the proposed system to the Cybex reference equipment used in the hospital.

### 4.4. Results from TKR Subjects

Eight TKR subjects aged 60–85 years participated in the experiments. They were instructed to perform the same experiments as the healthy control subjects. For each angular speed, 80 swing angles should have been received from the eight TKR subjects. After the removal of nine abnormal data, 71 swing angles remained available at each angular speed. The ROM is plotted in Figure 16a–c for the three speeds, respectively. The average absolute ROM errors with respect to the Cybex reference equipment were 1.65°, 2.74°, and 3.27° at the three angular speeds, respectively, as shown in Table 6. Compared with the healthy subjects, the accuracies for the TKR subjects were similar but the average absolute swing errors were smaller. This may be because they had used the equipment before, and thus they were more relaxed during the experiments and did not exert as much counterforce during the swing.

Correlation coefficients between the two measurement systems under three angular speeds shown in Figure 15 were calculated to be 0.993, 0.982, and 0.986, respectively. Again, they implied high correlation between the proposed system to the Cybex reference equipment.

After completing the rehabilitation on Cybex, each TKR subject sat on the chair to flex the leg back and forth approximately 90° 20 times. This was designed to simulate the rehabilitation scenario at home. The results showed no error in counting the swings from all subjects.

## 5. Conclusions and Future Work

Monitoring whether TKR patients are rehabilitated after the surgery remains a major concern for orthopedists. Without continuing rehabilitation, full recovery is delayed and the weak knee joint may affect the mobility of the patients, resulting in an urgent need for new devices or methods to overcome these problems. In this paper, an effective method is proposed to resolve the problems using three approaches: (1) monitoring whether the TKR patients follow the rehabilitation instructions at home, (2) automatically recording the duration of the rehabilitation course, and (3) saving the flexion angles and monitoring the progress from each rehabilitation course.

The proposed sensor device has social benefits and advantages such as usability without spatiotemporal constraints, reduction of frequency returning to the hospital for inpatient services, saving medical expenses, and accuracy in monitoring the rehabilitation progress. The developed sensor devices can be easily worn on the thigh and ankle, and the proposed method can calculate the number of swings and the equivalent ROM from each rehabilitation course. This fulfills the second and third goals of this study. An app was designed to display the swing angles so that users can track the effect of rehabilitation. The orthopedist can also monitor the progress of rehabilitation, thereby fulfilling the first goal of this study. The experimental results show that the average absolute swing errors from the TKR subjects were between 1.65° and 3.27° and that the accuracies were between 98.09% and 96.16% at different angular speeds.

Although the developed sensor devices are small and lightweight, they must be placed into a shell, and Velcro and an elastic belt are required to wear them on the leg. The hardware is proposed to be modified into a chip in the future. Furthermore, the developed devices are under investigation for other medical applications, such as rehabilitation for frozen shoulder, measuring trembling in Parkinson’s disease, identifying gait and joint patterns during walking, evaluating medial/lateral load or possible excessive stress shielding growth, and sport applications such as pitching pattern identification and adjustment.

## Figures and Tables

**Figure 1 sensors-20-01703-f001:**
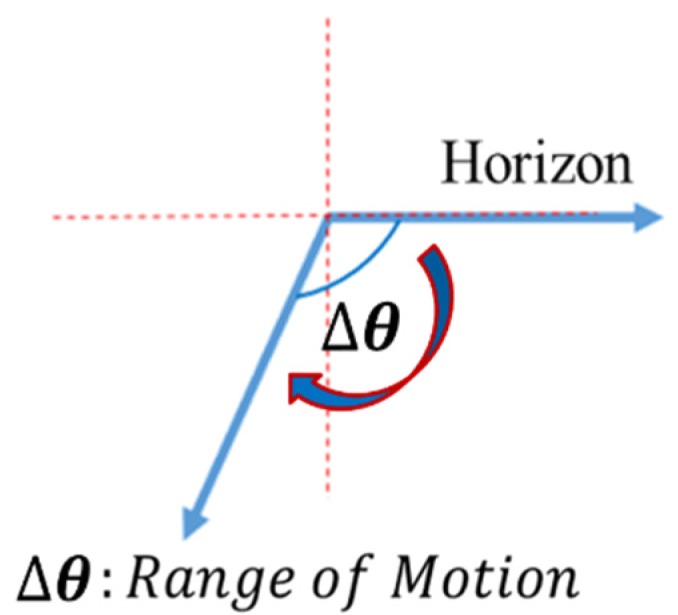
Range of motion (ROM).

**Figure 2 sensors-20-01703-f002:**
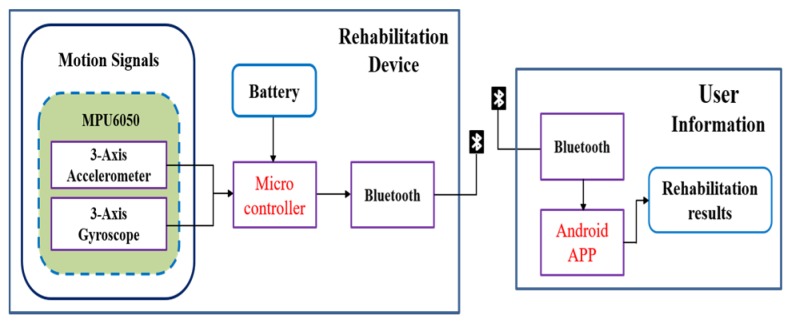
Structure of the proposed system.

**Figure 3 sensors-20-01703-f003:**
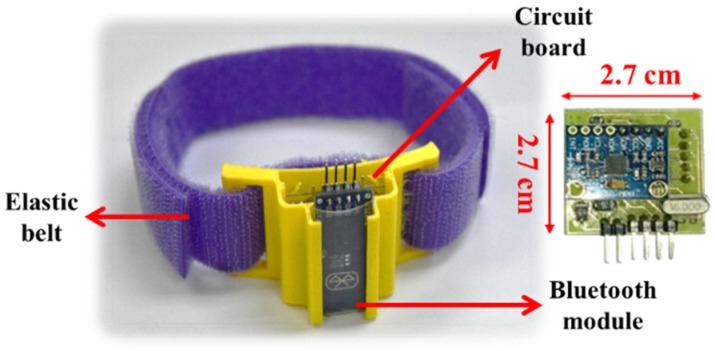
Developed sensor device.

**Figure 4 sensors-20-01703-f004:**
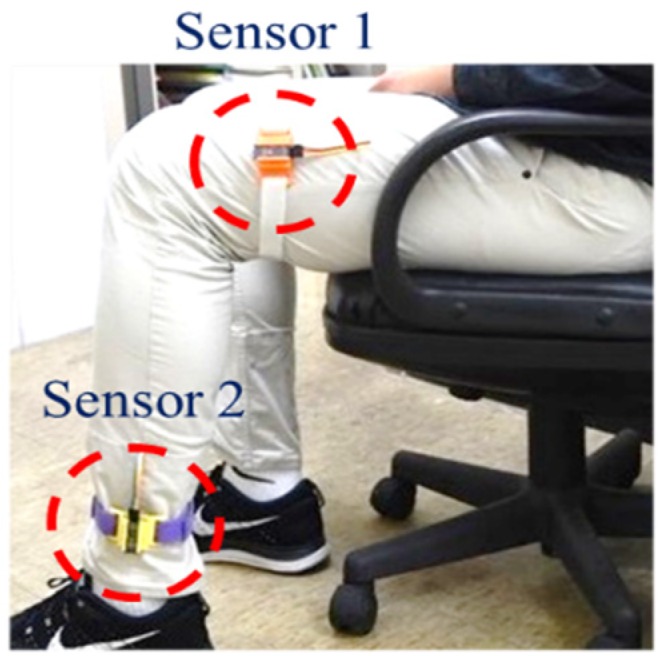
Two sensors worn on the thigh and ankle.

**Figure 5 sensors-20-01703-f005:**
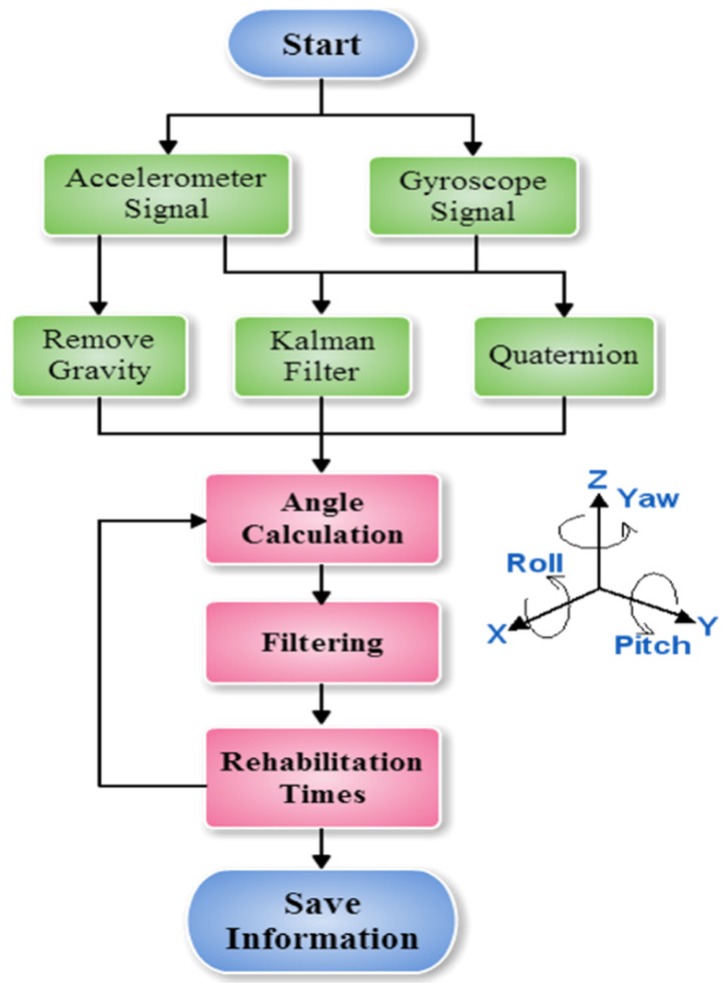
Proposed system flowchart.

**Figure 6 sensors-20-01703-f006:**
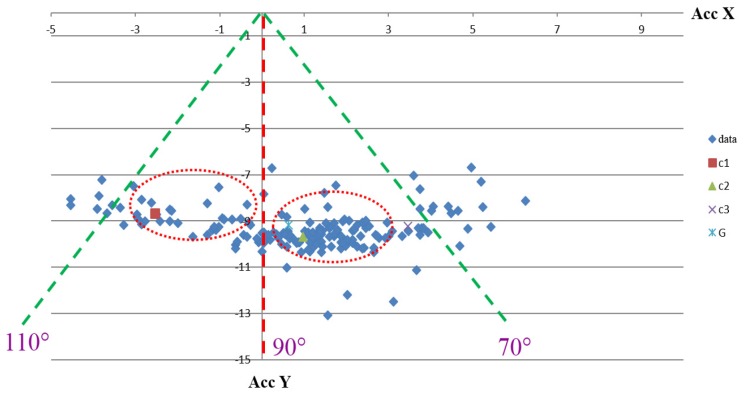
Detected angles affected by gravity at different swing velocities. Note that Acc X (unit: g) and Acc Y (unit: g) represent the measured acceleration from x- and y-direction of the accelerometer, respectively.

**Figure 7 sensors-20-01703-f007:**
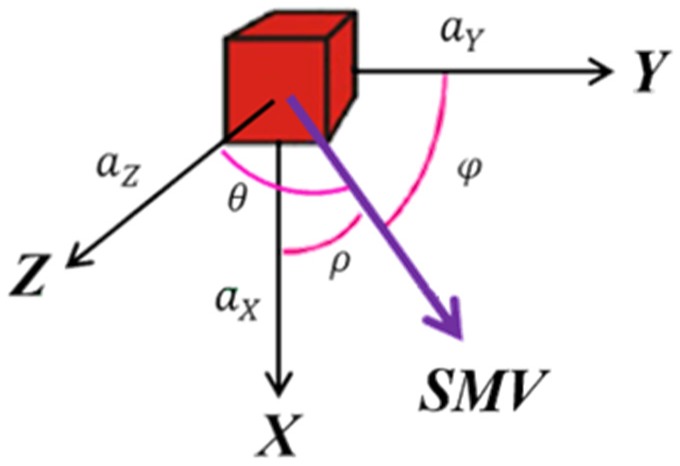
Angles between the signal magnitude vector (SMV) and the three axes.

**Figure 8 sensors-20-01703-f008:**
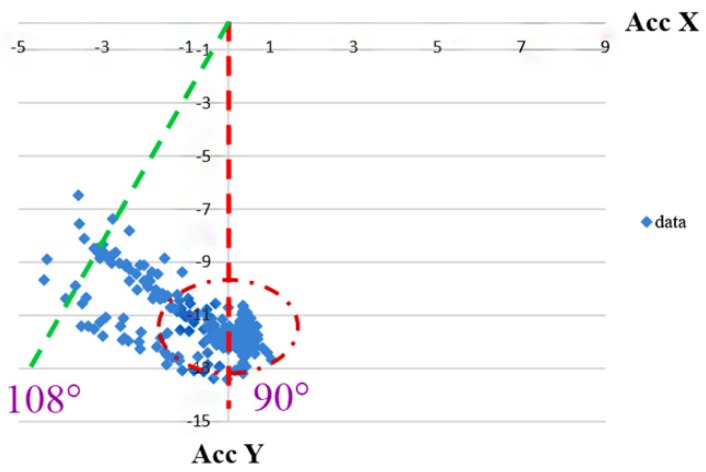
Detected angles after removing the gravity at different swing speeds. Note that Acc *X* (unit: g) and Acc *Y* (unit: g) represent the measured acceleration from *x*- and *y*-direction of the accelerometer, respectively.

**Figure 9 sensors-20-01703-f009:**
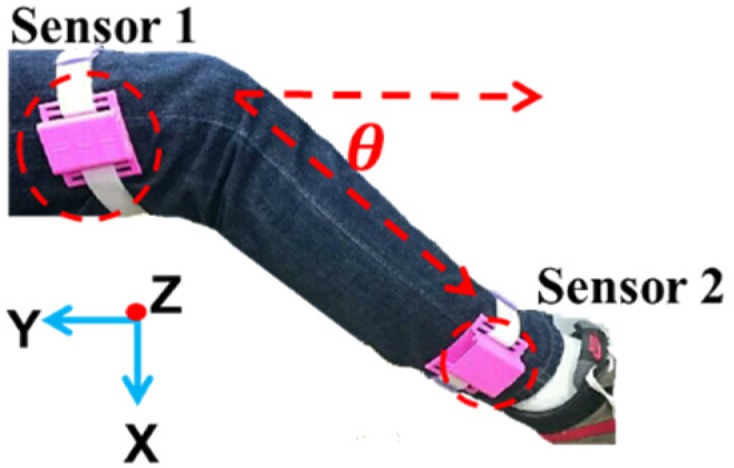
Equivalent angle of knee motion.

**Figure 10 sensors-20-01703-f010:**
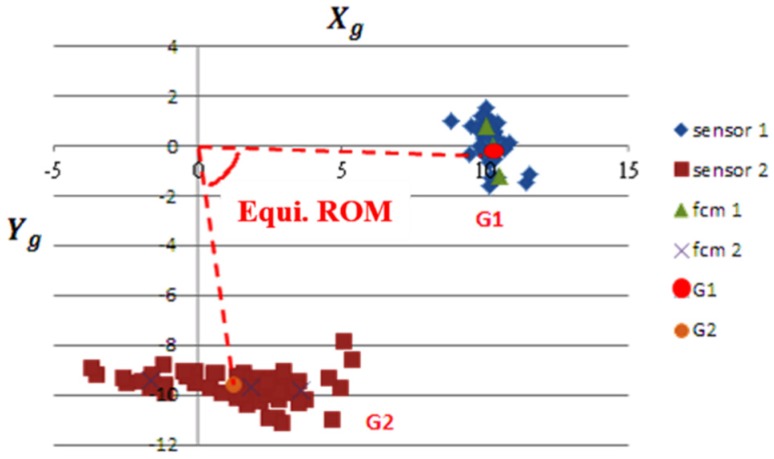
Equivalent ROM.

**Figure 11 sensors-20-01703-f011:**
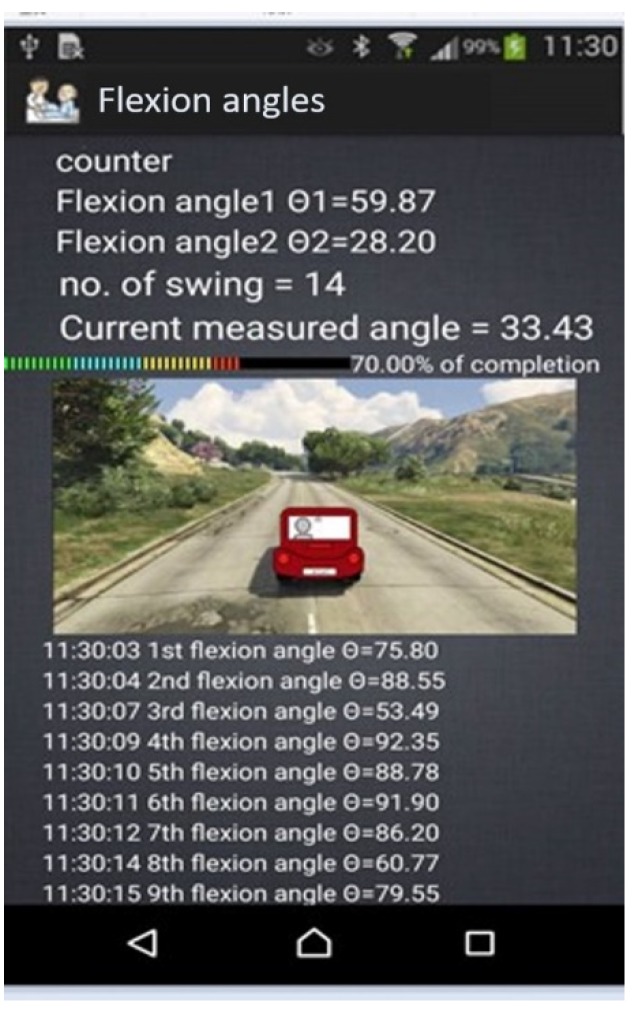
Rehabilitation interface.

**Figure 12 sensors-20-01703-f012:**
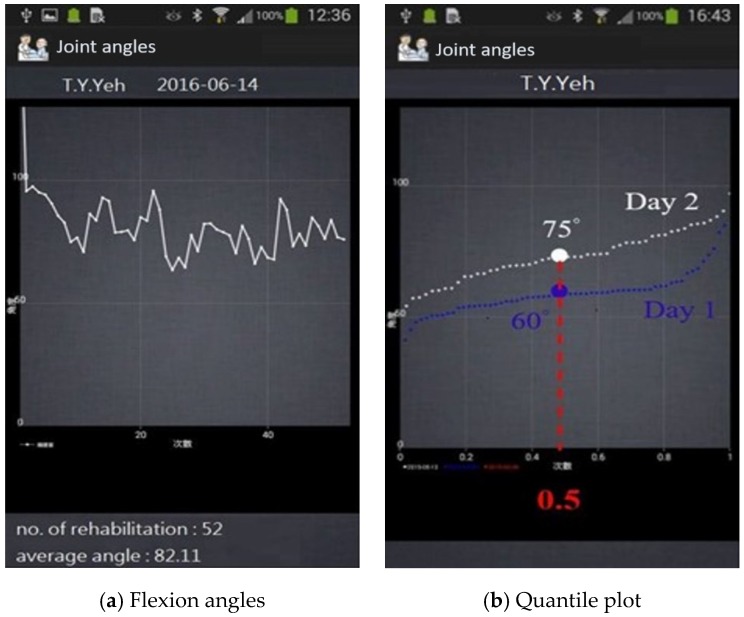
Comparisons of swing progress in quantile plots.

**Figure 13 sensors-20-01703-f013:**
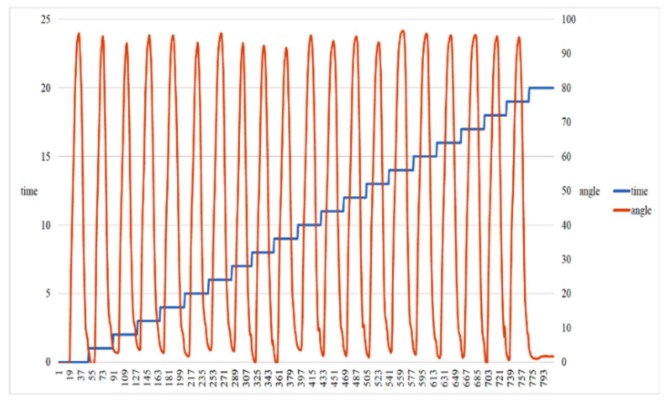
Swing angles and number of swings from one of the participants. Note that the horizontal axis represents the index of sequentially collected sensor signals.

**Figure 14 sensors-20-01703-f014:**
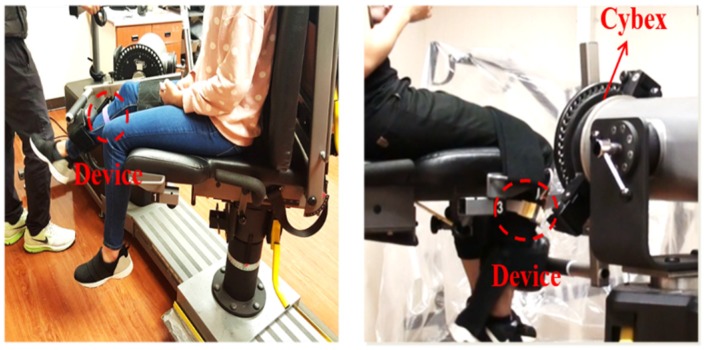
Experimental setup.

**Figure 15 sensors-20-01703-f015:**
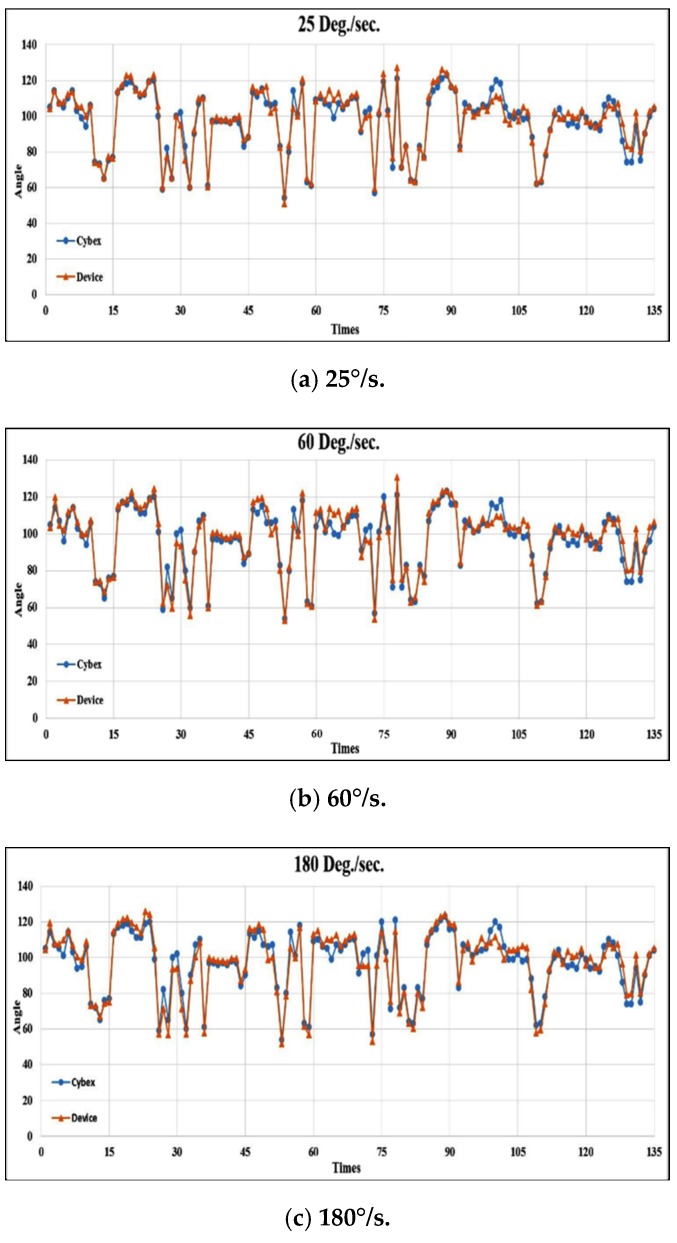
Comparisons of ROM for healthy control subjects at three speeds.

**Figure 16 sensors-20-01703-f016:**
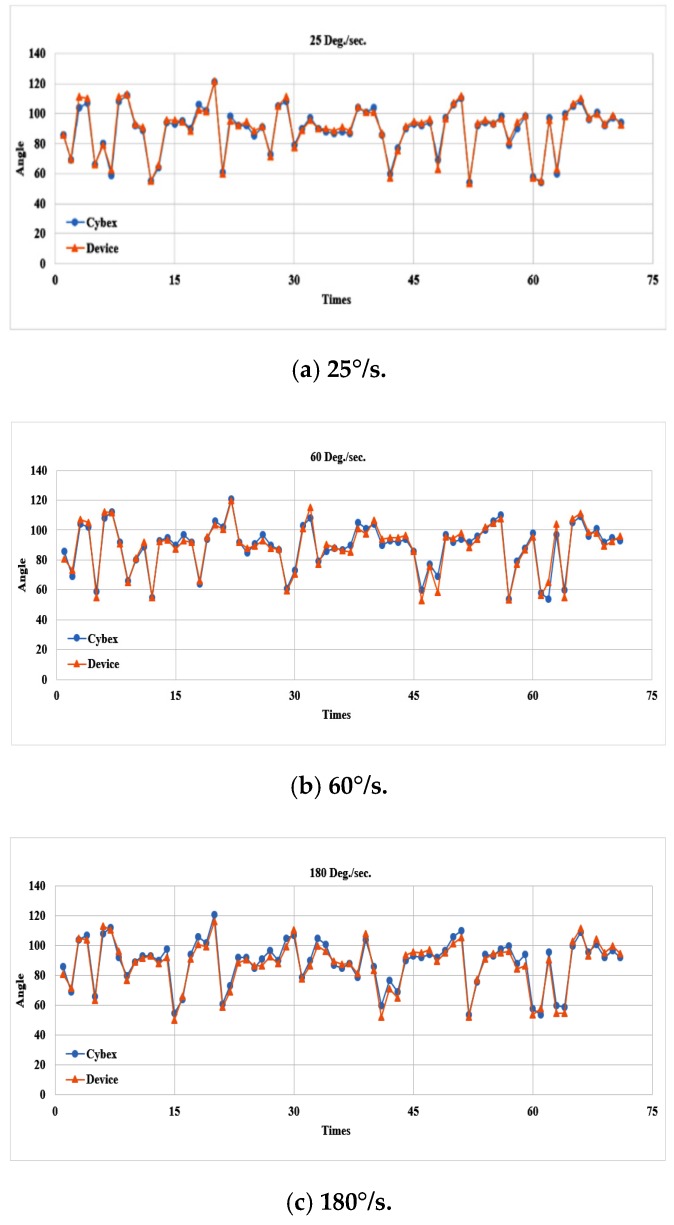
Comparisons of ROM for TKR subjects at three speeds.

**Table 1 sensors-20-01703-t001:** Specifications of the developed sensor device.

Size	33 mm × 32.5 mm × 16 mm
MCU	ATMEGA328P
IMU	GY-521(MPU6050)
Bluetooth	Arduino HC06
Battery	Lithium battery 7.4 V
Endurance	17.5 h

**Table 2 sensors-20-01703-t002:** Eleven subjects’ characteristics.

Gender	M	F
Number	7	4
Age (years)	33.71 ± 13.50	42.25 ± 12.97
Height (cm)	172.29 ± 8.10	154.50 ± 4.33
Weight (Kg)	70.29 ± 7.83	53.00 ± 1.87
Knee	Normal	Normal

**Table 3 sensors-20-01703-t003:** Healthy control subjects’ characteristics.

Gender	M	F
Number	8	8
Age (years)	64 ± 9.97	32.13 ± 16.06
Height (cm)	162.13 ± 2.52	159.38 ± 3.28
Weight (Kg)	68.39 ± 4.33	54.5 ± 4.15
Knee	Normal	Normal

**Table 4 sensors-20-01703-t004:** Total knee replacement (TKR) subjects’ characteristics.

Gender	M	F
Number	2	6
Age (years)	63.00 ± 3.00	70.67 ± 8.24
Height (cm)	161.70 ± 5.7	150.82 ± 3.75
Weight (Kg)	79.10 ± 9.10	57.43 ± 6.89
TKR (month)	7.50 ± 4.50	24.17 ± 16.80

**Table 5 sensors-20-01703-t005:** Average errors for healthy control subjects at three angular speeds.

Cybex	25°/s	60°/s	180°/s
Average absolute error (degree)	2.90	3.51	4.00
Standard Deviation (degree)	2.57	2.67	2.49
Accuracy (%)	97.01	96.31	95.77

**Table 6 sensors-20-01703-t006:** Average errors for TKR subjects at three angular speeds.

Cybex	25°/s	60°/s	180°/s
Average absolute error (degree)	1.65	2.74	3.27
Standard Deviation (degree)	1.29	2.05	1.54
Average accuracy (%)	98.09	96.71	96.16

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
