# Peer review of "Monitoring and Assessment of Rehabilitation Progress on Range of Motion After Total Knee Replacement by Sensor-Based System"

_sensors, 2020, doi:10.3390/s20061703_

Round 1
Reviewer 1 Report
Section 2 (in particular 2.2 and 2.3) are completely unclear to this reviewer. In particular: (with reference to fig 5) why gravity affects the measurament? is it a simple error in calibration for the IMU positioning (like it seems to be in fig 6) or the effective (excluded g) acceleration is used to reconstruct the knee flexion angle? what does the kalman filter do? it is used as usual to remove the "dalambertian acceleration" and compute the inclination of g with respect the frame fixed with the IMU, or a novel method in introduced? relation (4) computes the euler angles using the real acceleration components or the projection of g on the axes of the IMU frame? I think that all this aspects must be exposed in a exhaustive way before evaluating the rest of the work.
Author Response
Response to Reviewer #1 Comments
Thank you for the valuable and insightful comments about our original submission. We have revised the paper in every possibility to make the paper in much better quality. The followings are the response to your comments.
Q1: Section 2 (in particular 2.2 and 2.3) are completely unclear to this reviewer. In particular: (with reference to fig 5) why gravity affects the measurament? is it a simple error in calibration for the IMU positioning (like it seems to be in fig 6) or the effective (excluded g) acceleration is used to reconstruct the knee flexion angle?
RESPONSE:
- Thank you for the comments. The measured values of all inertial measurement units (IMU) will be affected by gravity. This is a problem existed in the hardware design. The reason is that they have metal plates inside. Whenever an acceleration changed, the capacitance varied in accordance with the metal plate moving up, down, left or right to measure the acceleration change. Acceleration in each direction of a moving object will be affected by gravity. Not to mention, the gravity will cause the measurement errors. It is not a simple error in calibration for the IMU positioning. So, many literature, such as [27, 28, 29], have presented methods to remove the effect of gravity that in turn can adjust the inertial sensing measurement to be more accurate.
- After excluding the gravity, effective acceleration signals were used to reconstruct the knee flexion angle.
Q2: what does the kalman filter do? it is used as usual to remove the "dalambertian acceleration" and compute the inclination of g with respect the frame fixed with the IMU, or a novel method in introduced?
RESPONSE:
As mentioned in Q1, acceleration in each direction of a moving object will be affected by gravity. Kalman filter and Euler Angle projection were two of the commonly used methods in the literature [27, 28, 29]. Kalman filter was applied as usual to remove the gravity effect in this paper. So, in Lines 150 ~ 154 in Section 2.2, we mentioned “To obtain accurate swing angles, we apply the signals from both the accelerometer and gyroscope to a Kalman filter [27, 28] for smoothing and using the quaternion to calculate the angles. To quantify the rehabilitation angles in the equivalent ROM, the effect of gravity is removed from the accelerometer so that the real acceleration signals from rehabilitation can be obtained [29].”
Q3: relation (4) computes the euler angles using the real acceleration components or the projection of g on the axes of the IMU frame? I think that all this aspects must be exposed in a exhaustive way before evaluating the rest of the work.
RESPONSE:
Thank you for the comments. Eq.(4) was obtained from using the projection of g on the axes of the IMU. So, we mentioned in Lines 190~193 below Eq.(4): “The results from another experiment after removing the effect of gravity are shown in Fig. 8. The error was approximately 10°, which is a considerable improvement compared with that in Fig. 6.”

Reviewer 2 Report
Comments to authors:
First, the work is valuable, and the main idea is very clear. There is a problem, the authors present a solution to address it, the solution is tested experimentally, and conclusions are extracted. All very “clean”.
The paper presents a measurement system to monitor the rehabilitation progress of patients that were submitted to a total replacement knee (TRK) surgery. The system shall be able to extract objective metrics such as number of swings, maximum knee flexion angle, and the duration of practice. The system shall be able to work anywhere, anytime, without spatiotemporal constrains.
And, in my opinion, the system fulfills all these requirements or, at least, hast a strong potential to do it.
Strengths of the paper:
> The subject is pertinent and aligned with the journal.
> The main idea is clear, the article is well organized, the text is well written, symbols and tables are consistent, and figures look good (in general). The main aspect of the paper is good.
> The research team is strong as it includes people from different areas, not only Engineering. The inputs and the feedback from physicians, surgeons and physiotherapists are fundamental! I gladly see that three authors are from the Health sector.
> The experimental results are robust as they include a significant population of real patients, and real medical equipment (Cybex). This is a good evidence of cross-collaboration between Engineers and Physicians.
Weaknesses of the paper:
> Some parts of the text need to be corrected/ improved. Some examples:
- Line 44, «According to a hospital’s tracking report…»: Please add a reference to the report.
- Line 47, «…important concerns that concern…»: Please eliminate the redundancy.
- Line 68, «van der Straaten…»: Capital letter.
- Figures 6 and 8: What is the meaning of the axes? Is distance? What are the units?
- Figure 13: What are the units of the axes? What is the meaning of the horizontal axis? Use markers, not colors, to identify the traces.
- Line 339: What are the units of G1 and G2?
- Figure 15: The three pictures, a, b and c, are identical! This is very strange! Please explain.
> Sections 2.3 and 2.4 are unbalanced. Section 2.3 explains with good detail the extraction of the attitude angles from the accelerometer. Unfortunately, the same detail is not present in section 2.4. Section 2.4 should be extended to explain how the attitude angles extracted from the accelerometer are fused with the attitude angles extracted from the gyro. More details about the Kalman filter should be provided. The Kalman filter is implemented by the IMU’s firmware or is implemented by the mobile application? The quaternions appear before or after the Kalman filter? Please improve section 2.4.
> Line 239: The text refers to FCM (Fuzzy C-Means) but does not provide any insights of the algorithm. In my opinion, the paper should give a brief explanation of the algorithm, enough for the reader to have a clear idea of how it works.
> Line 239: According to the text, the FCM clusters signals into three groups, but figure 10 only refers to two groups (G1 and G2). Please explain this mismatch.
> Figure 10: As far as I understand, groups G1 and G2 refer to the maximum and minimum angles of figure 13. Am I right? The meaning of the groups should be explained more clearly.
> Please give a brief explanation of the Cybex equipment. The authors are familiar with the equipment, but most readers are not. Cybex is motorized?
> Lines 351-356: It should be interesting to evaluate the frequency response of the measurement system. For example, by applying a known sinusoidal mechanical excitation, record the range of measurement and plot it against the frequency.
> Figure 15: The variance of ROM is very high. I was expecting curves quite regular. Can you justify?
> Line 374: «Eight TKR subjects aged 60-85 years…» This range of ages is not present in table 4.
In conclusion:
In my opinion, the paper, as it is, does not meet the requirements to be published in the Sensors MDPI Journal. However, it is a valuable work that, with extra effort, will converge to a final publishable version.
Author Response
Response to Reviewer #2 Comments
Thank you for the valuable and insightful comments about our original submission. We have revised the paper in every possibility to make the paper in much better quality. The followings are the response to your comments.
Q1: First, the work is valuable, and the main idea is very clear. There is a problem, the authors present a solution to address it, the solution is tested experimentally, and conclusions are extracted. All very “clean”.
RESPONSE:
Thank you.
Q2: The paper presents a measurement system to monitor the rehabilitation progress of patients that were submitted to a total replacement knee (TRK) surgery. The system shall be able to extract objective metrics such as number of swings, maximum knee flexion angle, and the duration of practice. The system shall be able to work anywhere, anytime, without spatiotemporal constrains.
RESPONSE:
Thank you. The proposed system can be used to work anywhere, anytime, without spatiotemporal constrains.
Q3: And, in my opinion, the system fulfills all these requirements or, at least, hast a strong potential to do it.
RESPONSE:
Thank you.
Q4: Strengths of the paper:
The subject is pertinent and aligned with the journal.
The main idea is clear, the article is well organized, the text is well written, symbols and tables are consistent, and figures look good (in general). The main aspect of the paper is good.
The research team is strong as it includes people from different areas, not only Engineering. The inputs and the feedback from physicians, surgeons and physiotherapists are fundamental! I gladly see that three authors are from the Health sector.
The experimental results are robust as they include a significant population of real patients, and real medical equipment (Cybex). This is a good evidence of cross-collaboration between Engineers and Physicians.
RESPONSE:
Thank you.
Q5: Weaknesses of the paper:
Some parts of the text need to be corrected/ improved. Some examples:
- Line 44, «According to a hospital’s tracking report…»: Please add a reference to the report.
RESPONSE:
Thank you for the comment. In our original expression, we did add reference [4] at the end of the sentence, “According to a hospital’s tracking report, some patients after TKR had knees that remained swollen or even deteriorated when the patients returned to the hospital for inpatient services [4].” To avoid confusion, we have corrected the expression to “According to a hospital’s tracking report [4], some patients after TKR had knees that remained swollen or even deteriorated when the patients returned to the hospital for inpatient services.”
- Line 47, «…important concerns that concern…»: Please eliminate the redundancy.
RESPONSE:
We have corrected the expression to “…are important clinical issues that concern orthopedists [6].”
- Line 68, «van der Straaten…»: Capital letter.
RESPONSE:
We have corrected it to “Van Der Straaten…”.
- Figures 6 and 8: What is the meaning of the axes? Is distance? What are the units?
RESPONSE:
The developed sensor was used to measure acceleration from the triaxial accelerometer. So, the unit is gravity, g = 9.8 m/s2. We have supplemented the expression in the caption of Fig. 6 and Fig. 8 by “Note that Acc X (unit: g) and Acc Y (unit: g) represent the measured acceleration from x- and y-direction of the accelerometer, respectively.”
- Figure 13: What are the units of the axes? What is the meaning of the horizontal axis? Use markers, not colors, to identify the traces.
RESPONSE:
As we mentioned in lines 305~308, “Each participant sat on a chair and flexed the leg back and forth at approximately 90° 20 times. The swing angles and number of swings from one of the participants are shown in Fig. 13.” The left vertical axis is time that represents the number of swings while right vertical axis is angle that represents the swing angle each time from the participant. Orange and blue colors were used to differentiate swing angles and number of swings, respectively. Furthermore, we have supplemented “Note that the horizontal axis represents the index of sequentially collected sensor signal.” to the caption of Fig. 13.
- Line 339: What are the units of G1 and G2?
RESPONSE:
G1 and G2 represent the centroid from each sensor device. So, their units are gravity, g.
- Figure 15: The three pictures, a, b and c, are identical! This is very strange! Please explain.
RESPONSE:
Although they may look similar, they are not identical. Three figures showed experimental results from three different swing speeds. There were many differences among them.
Sections 2.3 and 2.4 are unbalanced. Section 2.3 explains with good detail the extraction of the attitude angles from the accelerometer. Unfortunately, the same detail is not present in section 2.4. Section 2.4 should be extended to explain how the attitude angles extracted from the accelerometer are fused with the attitude angles extracted from the gyro. More details about the Kalman filter should be provided. The Kalman filter is implemented by the IMU’s firmware or is implemented by the mobile application? The quaternions appear before or after the Kalman filter? Please improve section 2.4.
RESPONSE:
- In lines 149~153 in Section 2.2, we mentioned “To obtain accurate swing angles, we apply the signals from both the accelerometer and gyroscope to a Kalman filter [27, 28] for smoothing and using the quaternion to calculate the angles. To quantify the rehabilitation angles in the equivalent ROM, the effect of gravity is removed from the accelerometer so that the real acceleration signals from rehabilitation can be obtained [29].” So, section 2.3 presented the effect of gravity on angle measurement.
- The system flowchart is presented in Fig. 5. In the flowchart, gravity removal, Kalman filter, and quaternion were performed before calculating swing angles. Section 2.4 is focused on the calculation of knee angle. So, in lines 205~207, we mentioned “We performed Kalman filtering on the rotation angle calculated from the acceleration and the angle integrated from the gyroscope angular velocity to determine the real roll and pitch angles.” So, the Kalman filter is implemented by the IMU’s firmware, not the mobile application.
- In lines 215~217, we mentioned “When applied to the six-axis inertia module, the quaternion can be calculated from the acceleration and angular velocity. This study used the MPU6050 inertia sensing module, which includes the quaternion in its functional library.” The quaternion operation appeared after we applied the Kalman filter to smooth the signals.
Line 239: The text refers to FCM (Fuzzy C-Means) but does not provide any insights of the algorithm. In my opinion, the paper should give a brief explanation of the algorithm, enough for the reader to have a clear idea of how it works.
RESPONSE:
We have supplemented a new paragraph in lines 228~232 in Section 3.1, “FCM is one of the commonly used machine learning methods that can softly partition data into the predetermined number of clusters [31]. A datum can be classified into any of the clusters with a membership degree between 0 and 1 under the constraint that the sum of membership degrees should be equal to 1. FCM was applied to calculate the equivalent ROM for the swing angles of rehabilitation.” Furthermore, a new reference [31] was added to the text.
Line 239: According to the text, the FCM clusters signals into three groups, but figure 10 only refers to two groups (G1 and G2). Please explain this mismatch.
RESPONSE:
As presented in Lines 241~250, we use FCM to cluster the signals from each sensor into three groups as indicated in the Step 2. Then, in Step 3, we find the centroid from each sensor device and represent the pair as (G1, G2). It means G1 and G2 represent the centroid of three clusters in sensor 1 and sensor 2, respectively.
Figure 10: As far as I understand, groups G1 and G2 refer to the maximum and minimum angles of figure 13. Am I right? The meaning of the groups should be explained more clearly.
RESPONSE:
No. G1 represents the centroid of three cluster centers obtained by FCM from sensor 1. Similarly, G2 is the centroid of three cluster centers obtained by FCM from sensor 2. In Lines 249~250 (Step 4), we perform basic operations on inverse trigonometric functions to calculate the equivalent ROM from the centroid pair (G1, G2). So, the centroid pair (G1, G2) was used to calculate the maximum and minimum angles in Fig. 13, not referring to maximum and minimum angles.
Please give a brief explanation of the Cybex equipment. The authors are familiar with the equipment, but most readers are not. Cybex is motorized?
RESPONSE:
- Yes, Cybex isokinetic dynamometer (or Cybex as shown in Fig. 14) is a motorized equipment and its swing speed can be manually adjusted to meet the user’s need. Cybex equipment was used as a benchmark for validating the performance of the proposed system.
- We have revised the expression in Lines 315~319 in Section 4.2, “Cybex is a motorized equipment used in the hospital and its swing speed can be manually adjusted to meet the user’s need. Experiments were then performed using professional rehabilitation equipment, Cybex, as a benchmark. To validate the effectiveness of the proposed system, only one sensor had to be worn at the shank during swings driven by Cybex, as shown in Fig. 14. The patient’s leg was bound tightly to the arm of Cybex so that the precise ROM could be recorded.”
Lines 351-356: It should be interesting to evaluate the frequency response of the measurement system. For example, by applying a known sinusoidal mechanical excitation, record the range of measurement and plot it against the frequency.
RESPONSE:
Thank you for the valuable comment. We will consider this interesting experiment in our future study. This paper is more focused on measuring patient’s swing angles and range of motion. As mentioned in Lines 305~308 in Section 4.1, “Two sensors were worn on the thigh and ankle, as shown in Fig. 4. Each participant sat on a chair and flexed the leg back and forth at approximately 90° 20 times. The swing angles and number of swings from one of the participants are shown in Fig. 13.” The proposed system can faithfully track the swing angles. Since the patient’s leg was not bound to the arm of Cybex so that the curves look like the sinusoid waves.
Figure 15: The variance of ROM is very high. I was expecting curves quite regular. Can you justify?
RESPONSE:
- The proposed measurement systems for healthy control subjects under three angular speeds, 25°/s, 60°/s, and 180°/s have standard deviations of 2.57, 2.67, and 2.49 degrees, respectively, as shown in Table 5. For TKR subjects, the standard deviations are 1.29, 2.05, and 1.54 degrees, respectively, as shown in Table 6. For the swing angles between 60~120 (or in average 90) degrees, no matter for healthy control subjects or for TKR subjects, the standard deviations were less than 3% which were not very high.
- Since the patient’s leg was bound tightly to the arm of Cybex so that the measured signals will not be so regular like the sinusoid waves. This is because if the Cybex was adjusted at a fixed speed such as 25°/s, its arm will have a random range of swing angles to help patient’s rehabilitation.
Line 374: «Eight TKR subjects aged 60-85 years…» This range of ages is not present in table 4.
RESPONSE:
Yes, eight TKR subjects aged 60–85 years participated in the experiments. But Table 4 shows the TKR subjects’ characteristics in which the age information was expressed by μ±s, where μ and s represent the mean and standard deviation of subjects’ age. We used the common expression as most literature did.
Q6: In conclusion:
In my opinion, the paper, as it is, does not meet the requirements to be published in the Sensors MDPI Journal. However, it is a valuable work that, with extra effort, will converge to a final publishable version.
RESPONSE:
We have tried in every possibility to revise the paper in much better quality. We sincerely hope the revised paper can be accepted for publication.
Round 2
Reviewer 1 Report
The authors satisfied all the request of this reviewer
Reviewer 2 Report
Nothing to add.